# Safety of Vaccines against SARS-CoV-2 among Polish Patients with Multiple Sclerosis Treated with Disease-Modifying Therapies

**DOI:** 10.3390/vaccines10050763

**Published:** 2022-05-12

**Authors:** Agata Czarnowska, Joanna Tarasiuk, Olga Zajkowska, Marcin Wnuk, Monika Marona, Klaudia Nowak, Agnieszka Słowik, Anna Jamroz-Wiśniewska, Konrad Rejdak, Beata Lech, Małgorzata Popiel, Iwona Rościszewska-Żukowska, Adam Perenc, Halina Bartosik-Psujek, Mariola Świderek-Matysiak, Małgorzata Siger, Agnieszka Ciach, Agata Walczak, Anna Jurewicz, Mariusz Stasiołek, Karolina Kania, Klara Dyczkowska, Alicja Kalinowska-Łyszczarz, Weronika Galus, Anna Walawska-Hrycek, Ewa Krzystanek, Justyna Chojdak-Łukasiewicz, Jakub Ubysz, Anna Pokryszko-Dragan, Katarzyna Kapica-Topczewska, Monika Chorąży, Marcin Bazylewicz, Anna Mirończuk, Joanna Kulikowska, Jan Kochanowicz, Marta Białek, Małgorzata Stolarz, Katarzyna Kubicka-Bączyk, Natalia Niedziela, Natalia Morawiec, Monika Adamczyk-Sowa, Aleksandra Podlecka-Piętowska, Monika Nojszewska, Beata Zakrzewska-Pniewska, Elżbieta Jasińska, Jacek Zaborski, Marta Milewska-Jędrzejczak, Jacek Zwiernik, Beata Zwiernik, Andrzej Potemkowski, Waldemar Brola, Alina Kułakowska

**Affiliations:** 1Department of Neurology, Medical University of Białystok, 15-276 Bialystok, Poland; amirtarasiuk@wp.pl (J.T.); katarzyna-kapica@wp.pl (K.K.-T.); chorazym@op.pl (M.C.); grandholy@gmail.com (M.B.); anna-mironczuk@wp.pl (A.M.); joannaakulikowska@gmail.com (J.K.); kochanowicz@vp.pl (J.K.); alakul@umb.edu.pl (A.K.); 2Faculty of Economic Sciences, University of Warsaw, 00-241 Warszawa, Poland; o.zajkowska@gmail.com; 3Department of Neurology, Jagiellonian University Medical College, 30-688 Krakow, Poland; marcin.wnuk@uj.edu.pl (M.W.); monika.marona@interia.pl (M.M.); claudianowak164@gmail.com (K.N.); agnieszka.slowik@uj.edu.pl (A.S.); 4Department of Neurology, Medical University of Lublin, 20-059 Lublin, Poland; annajamrozwisniewska@umlub.pl (A.J.-W.); krejdak@europe.com (K.R.); 5Neurology Clinic with Brain Stroke Sub-Unit, Clinical Hospital No. 2 in Rzeszow, 35-301 Rzeszow, Poland; fileminion@wp.pl (B.L.); gosiaur210474@interia.pl (M.P.); lidiaiadam.perenc@wp.pl (A.P.); 6Department of Neurology, Institute of Medical Sciences, Medical College of Rzeszow University, 35-310 Rzeszow, Poland; iwona.rosciszewska@op.pl (I.R.-Ż.); bartosikpsujek@op.pl (H.B.-P.); 7Department of Neurology, Medical University of Lodz, 90-419 Lodz, Poland; mariola.swiderek-matysiak@umed.lodz.pl (M.Ś.-M.); malgorzata.siger@umed.lodz.pl (M.S.); agnieszka.ciach@umed.lodz.pl (A.C.); agata.walczak@umed.lodz.pl (A.W.); anna.jurewicz@umed.lodz.pl (A.J.); mariusz.stasiolek@umed.lodz.pl (M.S.); 8Department of Neurology, Poznan University of Medical Sciences, 61-701 Poznan, Poland; karolina-kania@o2.pl; 9Faculty of Medicine, Poznan University of Medical Sciences, 61-701 Poznan, Poland; klara.dyczkowska@ump.edu.pl; 10Division of Neurochemistry and Neuropathology, Poznan University of Medical Sciences, 61-701 Poznan, Poland; akalinowskalyszczarz@ump.edu.pl; 11Department of Neurology, Faculty of Medical Sciences in Katowice, Medical University of Silesia, 40-055 Katowice, Poland; wgalus@sum.edu.pl (W.G.); awalawska-hrycek@sum.edu.pl (A.W.-H.); ekrzystanek@sum.edu.pl (E.K.); 12Department of Neurology, Wroclaw Medical University, 50-367 Wroclaw, Poland; justyna.ch.lukasiewicz@gmail.com (J.C.-Ł.); jaubysz@usk.wroc.pl (J.U.); anna.pokryszko-dragan@umw.edu.pl (A.P.-D.); 13Department of Neurology, Regional Specialised Hospital No. 4 in Bytom, 41-902 Bytom, Poland; marta_ewa@interia.pl (M.B.); s.neurologia@szpital4.bytom.pl (M.S.); 14Department of Neurology, Faculty of Medical Sciences in Zabrze, Medical University of Silesia in Katowice, 40-055 Katowice, Poland; kkubicka-baczyk@sum.edu.pl (K.K.-B.); natalia@niedziela.org (N.N.); nataliamorawiec007@gmail.com (N.M.); m.adamczyk.sowa@gmail.com (M.A.-S.); 15Department of Neurology, Medical University of Warsaw, 02-091 Warszawa, Poland; apodlecka@wum.edu.pl (A.P.-P.); monika.nojszewska@wum.edu.pl (M.N.); beata.zakrzewska-pniewska@wum.edu.pl (B.Z.-P.); 16Collegium Medicum, Jan Kochanowski University, 25-369 Kielce, Poland; ejasinska6@gmail.com (E.J.); wbrola@wp.pl (W.B.); 17Clinical Center, RESMEDICA, 25-726 Kielce, Poland; 18Department of Neurology and Neurorehabilitation, Miedzyleski Szpital Specjalistyczny, 04-749 Warszawa, Poland; jacekzaborski@icloud.com; 19Department of Neurology and Ischemic Strokes, Medical University of Lodz, 90-419 Lodz, Poland; milewskamarta88@gmail.com; 20Neurology Ward, Provincial Specialist Hospital, 10-561 Olsztyn, Poland; jacekzwiernik@gmail.com; 21Department of Neurology, University of Warmia and Mazury, 10-719 Olsztyn, Poland; zwiernik.beata@gmail.com; 22Clinic of Neurology, University of Warmia and Mazury, 10-719 Olsztyn, Poland; 23Department of Clinical Psychology and Psychoprophylaxis, University of Szczecin, 70-204 Szczecin, Poland; andrzej.potemkowski@wp.pl; 24Department of Neurology, Specialist Hospital in Końskie, 26-200 Końskie, Poland

**Keywords:** disease-modifying therapies, multiple sclerosis, vaccination, adverse event, SARS-CoV-2

## Abstract

(1) Background: The present study aims to report the side effects of vaccination against coronavirus disease 2019 (COVID-19) among patients with multiple sclerosis (MS) who were being treated with disease-modifying therapies (DMTs) in Poland. (2) Methods: The study included 2261 patients with MS who were being treated with DMTs, and who were vaccinated against COVID-19 in 16 Polish MS centers. The data collected were demographic information, specific MS characteristics, current DMTs, type of vaccine, side effects after vaccination, time of side-effect symptom onset and resolution, applied treatment, relapse occurrence, and incidence of COVID-19 after vaccination. The results were presented using maximum likelihood estimates of the odds ratio, *t*-test, Pearson’s chi-squared test, Fisher’s exact *p*, and logistic regression. The statistical analyses were performed using STATA 15 software. (3) Of the 2261 sampled patients, 1862 (82.4%) were vaccinated with nucleoside-modified messenger RNA (mRNA) vaccines. Mild symptoms after immunization, often after the first dose, were reported in 70.6% of individuals. Symptoms included arm pain (47.5% after the first dose and 38.7% after the second dose), fever/chills/flu-like symptoms (17.1% after the first dose and 20.5% after the second dose), and fatigue (10.3% after the first dose and 11.3% after the second dose). Only one individual presented with severe side effects (pro-thrombotic complications) after vaccination. None of the DMTs in the presented cohort were predisposed to the development of side effects. Nine patients (0.4%) had a SARS-CoV-2 infection confirmed despite vaccination. (4) Conclusions: Vaccination against SARS-CoV-2 is safe for people with MS who are being treated with DMTs. Most adverse events following vaccination are mild and the acute relapse incidence is low.

## 1. Introduction

Coronavirus disease 2019 (COVID-19) was declared a pandemic by the World Health Organization in March 2020 [1]. Since its outbreak, the disease has continued to spread globally. New variants of the severe acute respiratory syndrome coronavirus-2 (SARS-CoV-2) are discovered every couple of months. Several vaccinations against the virus are now available. The neutralizing antibody produced after vaccination appears to provide immune protection from symptomatic SARS-CoV-2 infection [2]. Patients with impaired immune systems (e.g., natural aging, neoplastic diseases, immunosuppressive treatment) are of high priority when vaccinating the population. However, these groups are usually not analyzed separately in clinical trials. Therefore, gathering real-world data is crucial. 

Vaccination against COVID-19 is recommended for patients with multiple sclerosis (MS) [3]. Early studies show the safety of immunization against COVID-19 among individuals with MS [4]. Polish recommendations for vaccination are consistent with international guidelines [5], owing to the risk of the severe clinical course of the infection and further complications that outweigh any potential risks from the vaccination [6]. Immunization was available for the general population in Poland from May 2021 (before that time, it was reserved for selected individuals according to their profession and age). However, some considerations are noted regarding the effectiveness of the immune response and the possible side effects of vaccination among patients with MS who are being treated with disease-modifying therapies (DMTs). Several DMTs were found to impair the humoral immune response to the vaccine [7,8]. In the literature, there are some case studies of MS relapses triggered by COVID-19 vaccination [9,10], but the relapse rate was not found to be significantly high in cohort studies [4]. Furthermore, the important issue in Poland is that the percentage of vaccinated individuals is very low, which shows insufficient confidence in the efficacy of immunization. 

The present study aimed to evaluate the safety of vaccination against COVID-19 among individuals with MS who were being treated with DMTs, based on the collected data from 19 Polish MS centers. The goal was to report any adverse events following vaccination and their impact on the disease course, including relapse. Such a large study has not previously been carried out among the population of Polish patients with MS.

## 2. Materials and Methods

### 2.1. Data Sources

The Multiple Sclerosis and Neuroimmunology Section of the Polish Neurological Society published an announcement regarding the current study at www.ptneuro.pl (accessed on 1 November 2021). Every MS Center in Poland was invited to participate, and participants were recruited from 16 Polish MS Centers. The data were gathered by neurologists using the same questionnaire for all MS Centers (see Appendix A). The interview was conducted with the patient once, after a year of availability of vaccination against COVID-19 in Poland.

Individuals diagnosed with MS according to the 2010 and 2017 McDonald criteria were included, and most of them were being treated with DMTs. The patients’ levels of disability were assessed using the Expanded Disability Status Scale (EDSS) [11]. Patient demographics data and the characteristics of MS were collected, which included the duration of the disease, the level of disability, current DMT use, information about vaccination against SARS-CoV-2 (type of vaccination, number of doses), the presence of adverse events following vaccination (time of onset and resolution, type and severity of symptoms, treatment used), and information regarding the occurrence of relapses after immunization or worsening of MS symptomatology (pseudo-relapse). The minimum observation time was 21 days. Gathered data included information about the number of doses (one or two) of different vaccines. The additional data about COVID-19 cases in the Polish population were obtained from official reports of the Polish Ministry of Health [12].

### 2.2. Timeline of Vaccination in Poland

From late December 2021, it was possible for health workers to register for the administration of BioNTech, the Pfizer vaccine. In the following months, registration options were extended to teachers, the uniformed services, selected age groups (starting with the oldest and successively including younger ones), and immunocompromised individuals (people with MS being treated with DMTs were not included). The Oxford, AstraZeneca vaccine and Moderna vaccine were available in Poland, with a slight delay, from February 2021. The first doses of Johnson & Johnson vaccines were included in the vaccination program from April 2021. The availability of different vaccine types varied among vaccination centers, especially during the first months of the national immunization program. In May 2021, it was possible for all adult persons to register for vaccination against COVID-19. A booster dose after the first full course of vaccination was available in Poland for the general population (including patients with MS) from November 2021. At the time of conducting the survey, the additional dose was recommended, but this was not included in the analysis as the time of observation was short. The basic timeline of vaccine availability in Poland is shown in Figure 1 [13].

### 2.3. Analyses

The χ^2^ test statistic of the homogeneity of odds was calculated to compare the reported side effects among the divided cohort due to different variables. Further analysis was controlled for several potential confounders, and logistic regression models were estimated. To assess the relevance of reported adverse events, we used Fisher’s exact test, Student’s *t*-test, the U Mann–Whitney test, and maximum likelihood estimates of the odds ratios. Given the space limitations and the aim of focusing on the most significant results, we skipped most of the estimation output tables or reduced the table’s dimensions. All analyses were performed using the STATA 15 software [14]. 

### 2.4. Standard Protocol Approvals

The study was approved by the Bioethics Committee at Collegium Medicum, Jan Kochanowski University, in Kielce, Poland (approval no. 62/2021).

## 3. Results

### 3.1. Descriptive Statistics of the Study Population

The study included information regarding 2261 individuals with MS who had been vaccinated against SARS-CoV-2. The data of 75 patients were excluded due to their non-eligibility for the study (a lack of crucial information). The follow-up time from the first dose of the vaccine ranged from 21 days to one year (mean 7 months).

Among those MS centers participating in the study, the percentage of patients with MS who were being treated with DMTs and were vaccinated against COVID-19 varied from 29.5% to 74.8%. Approximately 70.6% of all vaccinated patients reported at least one adverse event.

All patients were treated with different DMTs. The average age of all participants was 42.61 years old. Table 1 presents the demographics and clinical characteristic of the study group. Participants were treated with the following DMTs: dimethyl fumarate (891, 39.4%), interferon beta (518 individuals, 22.9%), teriflunomide (227, 10.0%), glatiramer acetate (209, 9.2%), natalizumab (131, 5.8%), fingolimod (109, 4.8%), ocrelizumab (91, 4.0%), cladribine (20, 0.9%), alemtuzumab (11, 0.5%), mitoxantrone (7, 0.3%), and other therapies (47, 2.1%).

In the study group, four vaccines were used: BioNTech, Pfizer vaccine (1684 patients; 74.5%), Oxford, AstraZeneca vaccine (264 patients; 11.7%), Moderna vaccine (178 patients; 7.9%), and Johnson & Johnson vaccine (135 patients; 6%). The first dose of the COVID-19 vaccine was administered to all patients and the second dose was administered to 2093 (92.6%) individuals. Most patients were given vaccines using mRNA technology (82.4%). In all patients, the same type of vaccine was used for the second as for the first dose. 

### 3.2. Timeline

The peak of vaccination frequency was observed between April and August 2021 in the general population, as well as among MS patients in Poland (Figure 2). Before that period, a major peak in the incidence of COVID-19 cases was observed in Poland (Figure 3).

### 3.3. Adverse Events

Mild symptoms after immunization, often after the first dose, were reported by 70.6% (1597) of individuals. The most common symptoms were flu-like symptoms (including fever and chills) and pain at the injection site. Table 2 presents the frequency of reported adverse events. In the cohort, 65.8% of males and 72.7% of females had at least one side effect following vaccination against COVID-19.

Only one severe adverse event was reported (pro-thrombotic complication in a 42-year-old female after receiving the AstraZeneca vaccine). Anaphylaxis occurred in 3 individuals. No fatal cases were registered. Nine patients were diagnosed with SARS-CoV-2 infection after vaccination, 6 after the first dose, and 3 after the second dose. 

### 3.4. Disease-Modifying Therapies

The percentage of patients with adverse events who were being treated with particular DMTs was as follows: 90.9% were treated with alemtuzumab, 85.7% were treated with mitoxantrone, 78.0% were treated with ocrelizumab, 74.1% were treated with natalizumab, 72.0% were treated with interferons, 71.0% were treated with dimethyl fumarate, 67.4% were treated with teriflunomide, 66.1% were treated with fingolimod, 63.6% were treated with glatiramer acetate, and 55% treated with cladribine. None of the DMTs were predisposed to the development of side effects, although the number of patients on cladribine, alemtuzumab, and mitoxantrone was insufficient to make a reliable inference. 

### 3.5. Adverse Events and Vaccine Type 

The individuals immunized with mRNA vaccines had a higher probability of having pain at the injection site in comparison to vaccines using non-replicating viral vectors (*p* = 0.017, OR = 1.419). Patients who received mRNA vaccines were less likely to develop other adverse events. We observed higher odds of developing headache and malaise after receiving a dose of the Moderna vaccine compared to the Pfizer vaccine (*p* = 0.0004, OR = 2.229; *p* = 0.0046, OR = 2.174). However, we must point out that the distribution of different vaccine types was unequal in our cohort so no definite conclusions can be drawn from the observed correlation. For reliable comparison, further evaluation is needed. 

### 3.6. Adverse Events and Age

The rate of fatigue associated with COVID-19 vaccination was higher in younger patients (*p* = 0.0227). Predisposition for younger patients was also found in the frequency of pain at the injection site and fever/chills/flu-like symptoms (*p* = 0.0031; *p* = 0.0000). Individuals with pain at the injection site after the first dose were on average 2.6 years younger compared to people without this adverse event. The difference was 2.3 years younger for people reporting this adverse event after the second dose. Headache was more commonly found in younger patients after the second dose (*p* = 0.0372). The associated symptoms were more frequent among individuals under 40 years of age (when dividing the cohort into groups for every 10 years of age). 

### 3.7. Adverse Events and Comorbid Diseases

Most of the comorbid diseases were not relevant to the adverse events experienced by individuals in the cohort. Only diabetes mellitus (present in 60 patients in the cohort) predisposed patients significantly to reported skin lesions at the injection site and fever after the first dose (*p* = 0.001, OR = 3.583; *p* = 0.008, OR = 2.162).

### 3.8. Adverse Events and EDSS

The value of EDSS among individuals with adverse events was lower in comparison to patients without side effects. However, the difference is close to the significance threshold and should be interpreted with caution (*p* = 0.049). Figure 4 shows the kernel density plot of EDSS distribution in the case of adverse events.

### 3.9. Adverse Events and Neurological Worsening

Relapse (a neurologic deficit associated with an acute inflammatory demyelinating event that lasts at least 24 h) after vaccination occurred in 99 (4.4%) individuals. In 16 (0.7%) patients, the first or second dose was administered less than 21 days before neurological worsening and in 83 (3.67%) patients beyond that time period. Clinical episodes that are classified as a relapse must have had a clear monophasic course, presented with objective findings that are typical of multiple sclerosis, must have lasted for over 24 h, and were not associated with fever or infection [9].

A temporary onset of worsening neurologic symptoms without clinical progression of the disease (pseudo-relapse) after vaccination was observed in 62 (2.7%) individuals. Those patients did not fulfill the criteria for relapse. Pseudo-relapse was present in 20 patients after the first dose and in 43 patients after the second dose. One individual had pseudo-relapse symptoms after receiving the first and second doses. In most cases, the symptoms appeared 1–2 days or over 8 days after the vaccination (regardless of the dose) (Figure 5). 

## 4. Discussion

This study provides a comprehensive analysis of Polish patients with MS who were being treated with DMTs, and who were vaccinated against SARS-CoV-2. Vaccination against SARS-CoV-2 among people with MS who were being treated with DMTs was found to be safe. The majority of patients from participating MS Centers were vaccinated with mRNA vaccines (Pfizer-BioNTech and Moderna), so our conclusions relate primarily to vaccination with this mechanism of action. These were first available for the population in Poland and preferred in patients with underlying medical conditions.

The most reported adverse events were pain at the injection site and fever/chills/flu-like symptoms. Localized arm pain and skin changes at the injection site were often reported after the first dose. However, other reported side effects were more frequent after the second dose. Fatigue and headache were less common among patients when compared with non-MS cohorts from other studies [15,16,17]. The symptoms were also similar in smaller study groups of patients with MS [4]. In our cohort, fever/chills/flu-like symptoms, fatigue, pain at the injection site, and headache (after the second dose) were more common in younger individuals. Natural aging of the immune system (decrease in humoral response) may lead to a different reaction to vaccination among specific age groups, which may be unrelated to MS [18]. Only one patient reported a serious adverse event, which was in the case of a 42-year-old female (RRMS, EDSS = 1, treated with interferon beta for 9 years) with no comorbid diseases, who had chest pain with an elevated D-dimer level 2 weeks after the first dose of the Astra Zeneca vaccine. Pulmonary embolism was excluded. However, the patient awaits cardiological consultation.

The profile of adverse events after vaccination against COVID-19 in our cohorts was consistent with those presented in previous studies that focused on individuals with other autoimmune disorders (e.g., autoimmune rheumatic diseases or inflammatory bowel disease) [19,20]. The frequency of adverse events was not higher when compared to that of the general population [15]. 

After vaccination against COVID-19, only 9 (0.4%) patients from the cohort were diagnosed with SARS-CoV-2 infection. The majority occurred before the second dose. None of the DMTs were dominant in this group, including the cell-depleting agents and fingolimod, which were found to impair the humoral immune response to the vaccine [7,8]. A long-term follow-up is required to better evaluate the protective value of COVID-19 vaccines.

The transient worsening of MS symptoms was reported in 62 (2.7%) individuals, often after the second dose. The percentage of patients experiencing pseudo-relapse after vaccination varies among studies, from 2% to 15% [4,21]. Interestingly, we observed transient neurological worsening with greater frequency after either a period of 1–2 days or over 8 days after vaccination. In the study by Lotan et al., most patients experienced neurological worsening during the first 24 h after vaccination. In this cohort, there was a downward trend in the number of patients reporting neurological worsening in the following days [21].

Within 21 days following vaccination, 16 (0.7%) patients presented with an acute relapse, while 83 (3.7%) patients experienced an acute relapse after 21 days. This relapse rate, especially during the first three weeks after vaccination, seems to be lower than the findings recorded by other studies [4,22]. However, no analysis was conducted in our study of non-vaccinated patients with MS during the corresponding time. 

Our study contains only adverse event data regarding the first and second vaccine doses against COVID-19 (mostly mRNA) since the length of observation time needed for monitoring patients after the booster dose and after full vaccination would be insufficient. The booster dose was available for individuals with MS and the general population in Poland from November 2021. After September 2021, the booster dose was only administered only to patients over 50 years of age and to immunocompromised patients (e.g., those undergoing oncological treatment or transplant recipients). 

The percentage of the Polish population who have been vaccinated against the SARS-CoV-2 virus is significantly lower than that of other European countries [23], and the mortality rate is one of the highest [24]. Regardless of the numerous studies and guidelines, there is still a severe lack of confidence in the safety and effectiveness of COVID-19 vaccines [25]. A great deal of misinformation being shared by social media increases public doubts about vaccination safety [26]. Providing patients with real-world data is crucial for reducing their hesitation regarding immunization. Therefore, our study argues that people with MS should be vaccinated, owing to the safety of vaccination against the SARS-CoV-2 virus among individuals with MS being treated with DMTs. 

The present study has several limitations. First, the study included only individuals with MS at the early stages of the disease (mean EDSS = 2), as most of this category of patients are treated with DMTs in Poland. Second, the study group was limited to patients who were reported by MS centers to be willing to participate in the research, and some individuals with MS were not reported. Third, the data were collected retrospectively; therefore, some individuals may have described their symptoms inaccurately. Fourth, the distribution of the different vaccines was unequal and our observations mainly relate to those vaccines based on the mRNA mechanism.

## 5. Conclusions

The outcome of our findings shows that vaccination against SARS-CoV-2 among people with MS who are being treated with DMTs is safe. Most adverse events are mild, and acute relapse incidence during the first three weeks after vaccination is low. Considering the current epidemiological situation, vaccination among individuals with MS should be promoted by neurologists and other healthcare workers.

## Figures and Tables

**Figure 1 vaccines-10-00763-f001:**
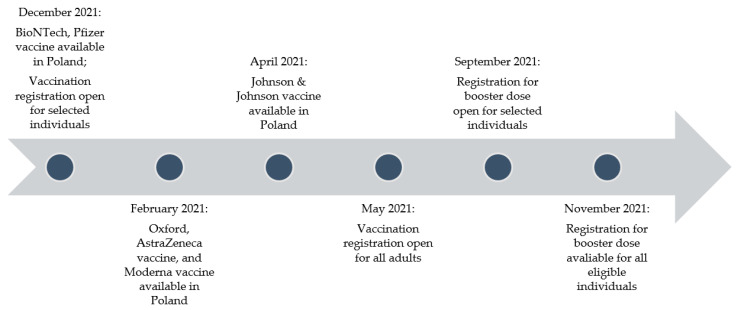
Timeline of vaccination and vaccine availability in Poland.

**Figure 2 vaccines-10-00763-f002:**
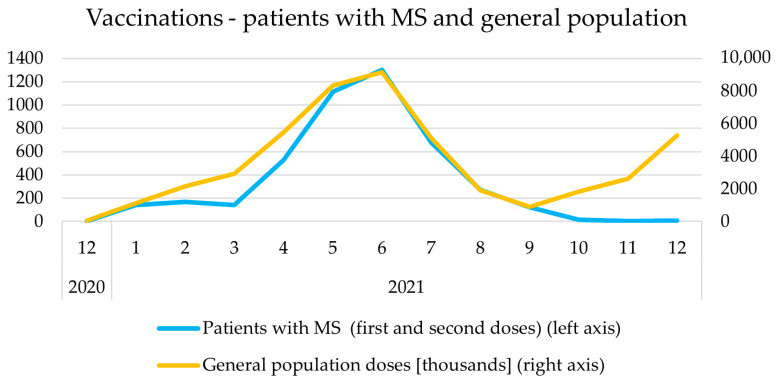
The vaccination frequency over the year in the cohort and in the general population in Poland.

**Figure 3 vaccines-10-00763-f003:**
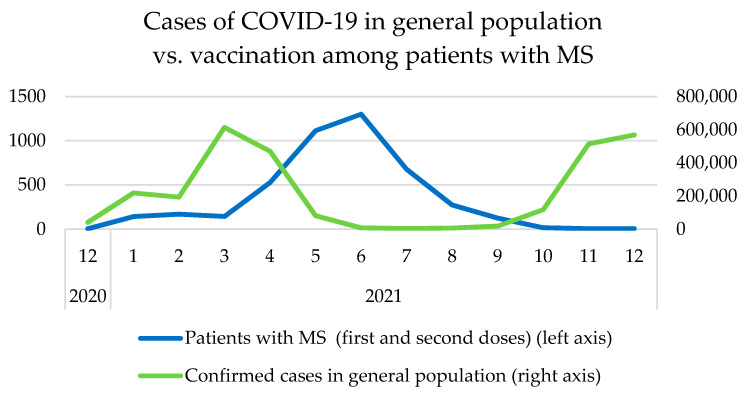
COVID-19 vaccination frequency among individuals with MS, in correlation with the confirmed cases in Poland.

**Figure 4 vaccines-10-00763-f004:**
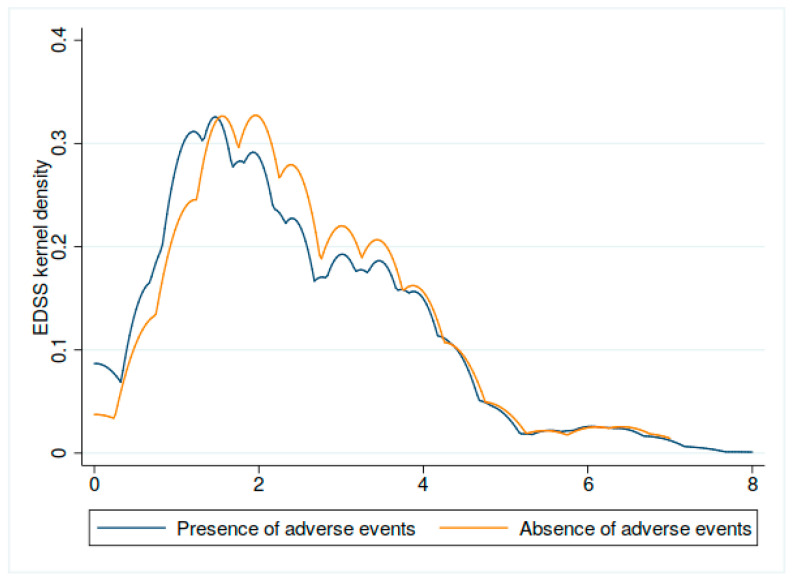
The kernel density plot regarding the EDSS distribution of adverse events among individuals with MS who were vaccinated against SARS-CoV-2 in Poland.

**Figure 5 vaccines-10-00763-f005:**
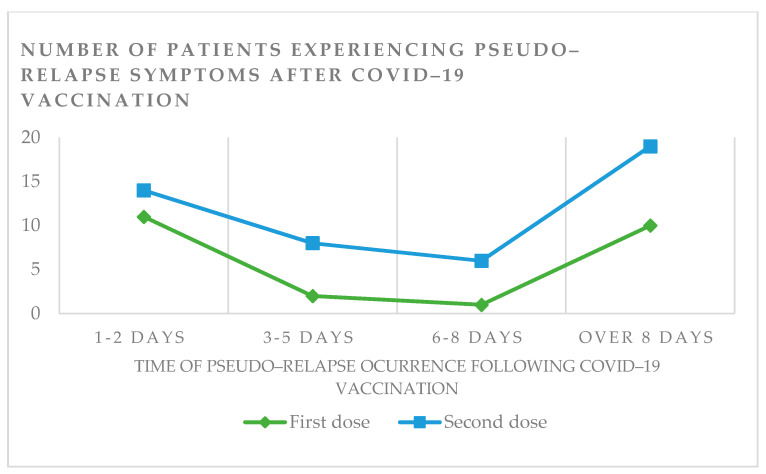
The incidence of pseudo-relapse in the cohort after the first and second doses of the COVID-19 vaccine, based on the onset of symptoms.

**Table 1 vaccines-10-00763-t001:** Demographics and clinical characteristics of patients with MS.

Demographics	No.	(%)	Range	Mean	MEDIAN	IQR	SD
Male	667	29.5					
Female	1594	70.5					
Male age			19–71	41.8	41	16	10.8
Female age			18–74	43	43	16	11.4
Clinical characteristics due to MS							
Disease course							
RRMS PPMS SPMS	2149 52 60	95.1 2.3 2.7					
EDSS			0–8	2.4	2	2	1.45
Disease duration (years)			0–41	9.48	8	8	6.35
Duration of DMT use (years)			0–21	5.64	5	5	3.82

Abbreviations: MS, multiple sclerosis; SD, standard deviation; IQR, interquartile range; RRMS, relapsing-remitting multiple sclerosis; PPMS, primary progressive multiple sclerosis; SPMS, secondary progressive multiple sclerosis; EDSS, the expanded disability status scale; DMT, disease-modifying therapy.

**Table 2 vaccines-10-00763-t002:** The incidence of mild adverse events, reported after the first and second doses of the COVID-19 vaccine.

	The First Dose of COVID-19 Vaccination		The Second Dose of COVID-19 Vaccination	
	No.	[%]	No.	[%]
Study population	2261		2093	
Pain at the injection site	1073	47.46	810	38.7
Skin changes around the injection site	106	4.69	85	4.06
Fever, chills, flu-like symptoms	387	17.12	428	20.45
Fatigue	233	10.31	236	11.28
Headache	206	9.11	198	9.46
Muscle pain/Joint pain	136	6.02	172	8.22
Diarrhea	2	0.09	2	0.1
Nausea/Vomiting	10	0.44	15	0.72
Abdominal pain	3	0.13	5	0.24
Malaise	133	5.88	172	8.22
Anaphylaxis	1	0.04	3	0.14

## Data Availability

The analysis was bundled into an open-R package accessible at https://github.com/Aczarnowska/COVID (accessed on 1 February 2022).

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
