# Peer review of "Safety of Vaccines against SARS-CoV-2 among Polish Patients with Multiple Sclerosis Treated with Disease-Modifying Therapies"

_vaccines, 2022, doi:10.3390/vaccines10050763_

Round 1
Reviewer 1 Report
Overall this is a well presented manuscript very much relevant to the literature required in this field.
Specific comments and suggestions from my review are:
Introduction should be improved to give a better background and relevance of the study. It should reflect summary of relevant literature review. Some of of the literature used in the discussion should be introduced here and then bring them back into contextual discussion with your results.
Other specific comments:
Line 66 - this claim is bit bold without the use of controls, needs to be supported.
Line 187 - what do you mean by other mechanisms - expand on this to clarify.
Lines 203-04 -not clear what is meant by this, please clarify the sentence.
Line 214 - relapse of what ? would be better to clarify.
Line 239 - compared with non MS cohorts?. I can't find mentioned in the methods. How or where did you get this data could be in the method.
Figure 4 - Y-xis missing and is also this figure is not clear to me.
Author Response
Response to Reviewer 1 Comments
Point 1:
Introduction should be improved to give a better background and relevance of the study. It should reflect summary of relevant literature review. Some of of the literature used in the discussion should be introduced here and then bring them back into contextual discussion with your results.
Response 1:Thank you very much for your valuable comment. We expanded the introduction.
Other specific comments:
Point 2:
Line 66 - this claim is bit bold without the use of controls, needs to be supported.
Response 2: Thank you very much for your valuable comment. To make it more clear, that the statement refers only to our study result we changed the sentence to:
“None of the DMTs in the presented cohort predisposed to the development of side effects.”
Point 3:
Line 187 - what do you mean by other mechanisms - expand on this to clarify.
Response 3: Thank you very much for your valuable comment. To make it more clear, we changed it to “non-replicating viral vectors”.
Point 4:
Lines 203-04 -not clear what is meant by this, please clarify the sentence.
Response 4: Thank you very much for your valuable comment. The construction of the sentence has been improved to provide more clarity.
“Most of the comorbid diseases were not relevant to adverse events experienced by individuals in the cohort.”
Point 5:
Line 214 - relapse of what ? would be better to clarify.
Response 5: Thank you very much for your valuable comment. We provided an additional explanation in brackets. To provide a very clear view of what was defined as relapse in our study the definition is provided in lines 215-218 of the originally send manuscript.
Point 6:
Line 239 - compared with non MS cohorts?. I can't find mentioned in the methods. How or where did you get this data could be in the method.
Response 6: Thank you very much for your valuable comment. We made a simple comparison with other previously published studies (cited in the discussion). A clarification was added.
Point 7:
Figure 4 - Y-xis missing and is also this figure is not clear to me.
Response 7: Thank you very much for your valuable comment. We modified figure 4 to provide more clarity.

Reviewer 2 Report
See attached

Author Response
Response to Reviewer 2 Comments
Point 1:
Introduction should include when the first vaccine was available for public use in Poland.
Response 1:Thank you very much for your valuable comment. The information is added and discussed in more detail in Material and Method section.
Point 2:
All percentages and numerical numbers with decimal points should be express only to the tenth (0.0%). Report the value to the 100th decimal provides no additional information, and the precision of your data does not justify this level of reporting.
Response 2: Thank you very much for your valuable comment. We have made the correction in the manuscript. We left it only in Table 2 as there is 1 patient with anaphylactic shock and after giving the value in decimal points it would be 0%. However, we can make the correction also in Table 2 if still recommended.
Point 3:
Was the same vaccine used for both vaccinations in all patients? If not, what percentage received a
vaccine from a different company for the second dose?
Response 3: Thank you very much for your valuable comment. The same vaccine was used in both doses in all patients. We added this important information for clarity.
Point 4:
Where did you obtain the information for Figure 1? Is the data regarding all MS patients in the Poland who were eligible to receive the vaccine? Based on the title it would appear that it is the entire population, but you need to provide more information in a footnote or other format identify how the information was obtained and from what source. Based on this figure It appears that there was no hesitancy in giving patients with MS the vaccine. Why was there a drop off between Sept and Dec 2021 for the MS population? Was everyone that was eligible with MS vaccinated?
Response 4: Thank you very much for your valuable comment. The information included in the manuscript are obtained from 16 MS Centers from Poland. The information was collected from MS Centers who were willing to participate (Matherials and Methods section). However, the final number of recruited patients is quite representative in comparison with other studies focused on patients with MS. In Poland the hesitancy towards vaccination is a major issue not only among the general population but also among individuals with MS (what we wanted to show in this figure). The percentage of vaccinated people in Poland is still very low. One of the reasons for hesitancy are concerns about vaccines safety. That is why we genuinely believe that our research can be a helpful tool while convincing unvaccinated ones.
The percentage of vaccinated patients among our MS Centers is still quite low and the biggest MS Centers participated:
“Among MS Centers participating in the study, the percentage of patients with MS, treated with DMTs, and vaccinated against COVID-19 varied from 29.5% to 74.8%. “ (citation from the manuscript)
Where the drop came from we can only speculate. One hypothesis is that at the beginning of vaccine availability, all the convinced ones got vaccinated and that the obstacle of convincing the hesitating ones came.
Point 5:
Where did you obtain the information for Figure 2? No surprise that vaccination rates would go up after increased incidence of the disease and increased availability of the vaccine. Not sure that this figure really adds anything to this paper.
Response 5: Thank you very much for your valuable comment. The data were obtained from official reports of the Polish Ministry of Health (Matherial and Method section , citation: “The additional data about COVID-19 cases in the Polish population were obtained [7].”). We wanted to put the vaccination rate among individuals with MS in the context of the incidence rate in Poland (as it was slightly different in different countries). It shows people with MS motivation for vaccination at the beginning. We really wanted to put it in the context of epidemiological situation in Poland. Therefore we would really appreciate the opportunity to keep the figure.
Point 6:
There is no discussion as to when the vaccine became available in Poland. That would help to
understand the delay in vaccinations.
Response 6:Thank you very much for your valuable comment. This issue is very important, therefore we added ‘2.2. Timeline of vaccination in Poland’ section to the Materials and Methods section.
Point 7:
Your numbers really do not allow for a good subgroup analysis, but the timeline of vaccine availability in Poland would help the reader understand the data better.
Response 7:Thank you very much for your valuable comment. This issue is very important, therefore we added ‘2.2. Timeline of vaccination in Poland’ section to the Materials and Methods section.
Ponit 8:
There is no discussion as to which vaccines were available at what time during the pandemic and this study. That is important because if might illustrate that your experience with some vaccines is more limited than others throughout the 12 month period and may also explain the difference in the percentage of patient received each of the vaccines. Right now, you are reporting adverse reactions in general with all vaccines pooled together with no information on whether there might be a difference between vaccines.
Response 8: Thank you very much for your valuable comment. This issue is very important, therefore we added ‘2.2. Timeline of vaccination in Poland’ section to the Materials and Methods section.
Differences that we were able to observe between the vaccines are included in “3.5. Adverse events and vaccine type” section of Results. However, as the distribution of different vaccines was unequal the relations are not so clear. We added additional statement to limitation section.
Point 9:
Methodology need to state how many times the interview was conducted with the patient. Was it after each dose or only once during the 12 month period?
Response 9: Thank you very much for your valuable comment. The interwiev was conducted once. We added the information in Material and Method section.
Point 10:
The majority of the patient were given the BioNTech / Pfizer vaccine. In section 3.5, there is a subgroup comparison of a few of the adverse reactions. The problem with the data regarding the other three vaccines is their sample size is much smaller than the BioNTech / Pfizer group and might underrepresenting the true incidence of the various adverse reactions. This issue is not addressed in section 3.5 and needs to be stated. It would also be helpful to include a comparison table, so the reader is aware of this problem.
Response 10: Thank you very much for your valuable comment. We added an additional comment to section 3.5 to emphasise the limitation.
Point 11:
Section 3.6 Define younger and older patients within the first sentence. Is 2-3 years difference really a meaningful difference? What was the incidence rate for those less than 40 years and those 40 years and older?
Response 11: Thank you very much for your valuable comment. We fully agree that in a clinical sense this small difference is not very meaningful. It is the correlation that we found and was statistically significant so we wanted to mention it as calculations according to age are usually made and need to be stated. As the differences were very little we described what we found without a major analysis focused only on age (which if required we can add). The difference is not very big, however consistent with some other studies focused on patients with MS (https://pubmed.ncbi.nlm.nih.gov/33856242/). Our cohort is not very variable according to age (because it includes only individuals treated with DMTs) :IQR=16 years, 75th percentile is 51 years, 90th percentile is 58 years, 95th percentile is 62 years
Point 12:
Section 3.7 How many patients had diabetes in the study population? Providing a comparison against this group without knowing these numbers makes the reported ORs meaningless.
Response 12: Thank you very much for your valuable comment. Among the cohort 60 patients had diabetes. We added this important information.
Point 13:
Section 3.8 There really is no such term as “barely significant”. If you rounded the number it would be 0.05. The reported p-value is for what parameter? The sentence simply states “However, the difference is barely significant (p=0.049).” No values or outcomes are provided, just a statement about the statistical significance.
Response 13: Thank you very much for your valuable comment. We fully agree that the wording is unfortunate and we modified it to be more accurate. (‘However, the difference is close to significance threshold and should be interpreted with caution (p = 0.049).’)
Since EDSS is an ordinal measure, we compare distributions and therefore we do not provide any particular number describing the difference (there is no particular number we can provide unfortunately, however the statement is true).
Point 14:
Discussion section There are numbers regarding worsening of MS symptoms and relapse rates that are not in the results section 3.9, but are included in the discussion section.
Response 14:
Line 260: 3.01%
Thank you very much for your valuable comment. The mistake has been corrected.
Line 267: 0.71%
Thank you very much for your valuable comment. It is now added in section 3.9.
Line 268: 3.67%
Thank you very much for your valuable comment. It is now added in section 3.9.
